# Urinary Proteomics of Simulated Firefighting Tasks and Its Relation to Fitness Parameters

**DOI:** 10.3390/ijerph182010618

**Published:** 2021-10-11

**Authors:** Ting Zhu, Yuxiang Hu, Jooyeon Hwang, Dan Zhao, Libin Huang, Liang Qiao, Ankui Wei, Xin Xu

**Affiliations:** 1School of Kinesiology, Shanghai University of Sport, Shanghai 200438, China; 2011517005@sus.edu.cn (T.Z.); yx.hu@bionovogene.com (Y.H.); 2Department of Occupational and Environmental Health, Hudson College of Public Health, University of Oklahoma Health Sciences Center, Oklahoma City, OK 73104, USA; Jooyeon-Hwang@ouhsc.edu; 3Department of Chemistry, Fudan University, Shanghai 200433, China; 21110220130@m.fudan.edu.cn (D.Z.); liang_qiao@fudan.edu.cn (L.Q.); 4Baoshan Fire and Rescue Division of Shanghai, Shanghai 201901, China; tasselzhang@aliyun.com; 5Shanghai Anti–Doping Laboratory, Shanghai University of Sport, Shanghai 200438, China

**Keywords:** firefighting, urine, proteomics, liquid chromatography–mass spectroscopy

## Abstract

Firefighting rescues are high-hazard activities accompanied by uncertainty, urgency, and complexity. Knowledge of the metabolic characteristics during firefighting rescues is of great value. The purpose of this study was to explore the firefighting-induced physiological responses in greater depth. The urine samples of ten firefighters were collected before and after the simulated firefighting, and the proteins in urine samples were identified by the liquid chromatography–mass spectroscopy. Blood lactate and heart rate were measured. There were 360 proteins up-regulated and 265 proteins downregulated after this simulated firefighting. Changes in protein expression were significantly related to acute inflammatory responses, immune responses, complement activation, and oxidative stress. Beta-2-microglobulin (r = 0.76, *p* < 0.05) and von Willebrand factors (r = 0.81, *p* < 0.01) were positively correlated with heart rate during simulated firefighting, and carbonic anhydrase 1 (r = 0.67, *p* < 0.05) were positively correlated with blood lactate after simulated firefighting. These results illustrated that Beta-2-microglobulin, von Willebrand, and carbonic anhydrase 1 could be regarded as important indicators to evaluate exercise intensity for firefighters.

## 1. Introduction

The occupational activities of firefighters expose them to high-hazard work situations, which are diverse, unpredictable, and complex [1,2]. Non-fire emergency, such as variable natural disasters, usually cause building collapse and induce wounds, cuts, and bleeding during firefighter activities [3]. Firefighters also encounter toxic smoke or gas inhalation and high radiant heat loads in firegrounds [4,5]. These disasters result in severe human casualties and economic losses [6]. National Fire Protection Association (NFPA) estimated that 60,825 firefighter injuries occurred in the United States in 2019 [3]. The total cost of fire in the United States in 2014 was USD 328.5 billion, which was 1.9% of the U.S. Gross Domestic Product [7]. According to China Fire Protection Yearbook, between 2016 and 2020, 11,253 people injuries and death occurred in China, and the property losses caused by fire was RMB 186.16 billion (USD 28.73 billion) [8].

When performing rescue tasks, firefighters wear heavy and cumbersome personal protective equipment (PPE) weighting approximately 22 kg, carrying additional 9 to 18 kg weighted rescue apparatus [9,10,11]. Firefighters often climb stairs and ladder, and lift and carry heavy objects [6]. Further, sometimes rescue operations of large-scale disasters are of longer duration, and result in large quantities of energy to be expended (about 80% of heart rate maximum during firefighting activities) [12]. The International Association of Firefighters (IAFF) proposes that candidate firefighters must perform the Candidate Physical Ability Test (CPAT), which is an integrated simulation firefighting task including eight rescue programs, and aims to help firefighting agencies measure the physical fitness of candidate firefighters through simulated rescue tasks [11].

Knowledge of the metabolic characteristics during realistic firefighting rescue is of great value for adjusting the training program selectively and for good recovery for high-intensity rescues, and it is subsequently beneficial for decreasing the rate of injuries of firefighters. Due to the emergency nature of rescue works, physiological variable to real firefighting is difficult to measure [13]. Therefore, many studies designed simulated firefighting exercises according to specific research aims and measured physiological responses and physical performance of firefighters during simulated rescue tasks [13,14,15]. These studies mainly concentrated on cardiorespiratory fitness and energy requirement or cost of a simulated intervention [16], and the heart rate [17], oxygen uptake [16,18], expiratory ventilation [19], blood lactate concentration [20], and rating of perceived exertion (RPE) [21] were usually measured. Although these evaluation indexes could reflect some specific physiological alterations in simulated firefighting tasks, it is difficult to give a full picture of how simulated rescues may affect human physiology.

Proteomics techniques developed rapidly and proteomic analysis methods have been increasingly used to provide insight into complex physiological responses associated with different physical activities [22,23]. The high-throughput capability of proteomics allows simultaneous examination and analysis of a large number of proteins, which provides information about all physiological processes [24]. Proteomics analysis could reveal changes in protein content and abundance in response to different exercises, and provide useful information for guidance of exercise training, nutrition, and immune function [23,25,26]. To the best of our knowledge, there are no proteomics analysis applied to the study of high-intensity rescues for firefighters.

Urine is an ideal source for exercise proteomic studies. Firstly, sampling of sufficient volume of urine is relatively easy. Secondly, compared with blood, urine contains less lipid and more polypeptides, and it has no homeostatic mechanism. Most importantly, urinary proteins have been served as noninvasive biomarkers monitoring the body’s stress under various physiological changes and identify the body’s condition in different exercise situations [26,27]. Most previous studies applied standard dipstick proteinuria method and specific assay for microprotein concentrations to monitor training load, physical performance and exercise capacity [28,29]. However, these methods could not reflect the physiological changes induced by exercise comprehensively, and related relative mechanisms could not be fully understood. Proteomics have the characteristics of high throughput, sensitivity, and quantitative, which could dynamically reveal the composition, function, and relation of all proteins under specific exercises from a holistic perspective [26]. Therefore, urinary proteomics could be applied for exploring the metabolic characteristics during simulated firefighting tasks.

Due to the emergency nature of rescue works, it is difficult to obtain urine samples during real firefighting. Therefore, it is more practical to measure physiological responses of simulated rescue tasks. This study designed simulated firefighting tasks according to the characteristics of real firefighting rescue combined with CPAT, and then explored the simulated tasks-induced physiological responses using the urine proteomics analysis. The study could provide a reference for assessing the physiological responses to high-intensity tasks, finding ideal biomarkers to monitor the body’s stress and then guarantying the safety of firefighters and adjusting the daily training plan.

## 2. Materials and Methods

### 2.1. Subjects

Subject recruitment was carried out in the Fengxian District, Shanghai, China, November 2020. To reduce the differences between subjects, the inclusion criteria were: (1) aged 18 years and older; (2) length working for the firefighting ≥1 year; (3) 18.5 ≤ BMI < 24; and (4) in good general health with no history of cardiovascular or respiratory disease.

### 2.2. Ethical Approval

The study was approved by the ethics committee of Shanghai University of Sport (No: 102772020RT106). Before the experiment started, all participants had detailed procedures introduced and signed the informed consent documents.

### 2.3. Simulated Firefighting Tasks

Referring to the relevant evaluation criteria and studies [11,14,15], this study designed five representative and specific activities of firefighting: (1) climb 6 flights of stairs; (2) lift heavy objects weighting 19 kg and walk downstairs; (3) pick up a 4.0-kg sledge hammer and strike the large tire weighting 72.5 kg to make it move forward 1.5 m; (4) after crossing the 22-m obstacle, drag a 44-mm hose equipped with a 3-kg automatic nozzle at a distance of 22.0 m to a pre-positioned drum, and then open the nozzle to hit the target; and (5) grasp an 80-kg manikin and drag it 30 m to the finish line. Throughout the five simulated tests, subjects wore a 22.7-kg vest to simulate the weight of the firefighter protective clothing ensemble and self-contained breathing apparatus, and carried a backpack air bottle (11.3 kg, 25–30 MPa).

However, the actual rescue tasks are complicated as they require strength, endurance, flexibility, anaerobic, and aerobic fitness, and contributions of each type physical fitness to the physiological responses caused by firefighting could not be well qualified only by the simulation firefighting test. Therefore, this study combined simulated firefighting tasks with different types of physical performance tests. By the combined approach in this study, it could help to identify factors that contributed the physiological responses to simulated firefighting more precisely by analyzing the correlations between physical fitness performance, simulating firefighting scores and urine proteomics (Appendix A).

### 2.4. Performance Tests

The performance tests included the pro-agility tests (5–10–5), one repetition maximum, the crunch tests, the sit-and-reach test, the 300-yard shuttle run, and the 2400-m run.

#### 2.4.1. Pro-Agility (5–10–5)

The participants started in a neutral stance, straddling the start line. On the “Go” command, the participants turned to left and ran 4.55 m (5 yd), touching the cone with their left hand. They then turned to the right, ran 9.10 m (10 yd) to the far cone, and then touched it with their right hand. Finally, they sprinted 4.55 m (5 yd) to the neutral line. Each underwent the test two times, and the best score was recorded.

#### 2.4.2. One Repetition Maximum

During the One Repetition Maximum test (bench press and deep squat), a warm-up set of 5–10 repetitions was performed. Then, participants performed 3–5 maximal trials, followed by an assessment of 1 RM strength. When the subject felt exhausted or failed to complete the trial, multi-RM was used to predict. In this study, the relative strength was calculated by dividing the absolute strength by the body weight.

#### 2.4.3. The Crunch Tests

The subjects lied in a supine position, bent knees and placed hands at both sides. The first line was at the position of fingers, and the second line was 12 cm apart from the first line. A metronome was set at 20 beats per minute. The test was concluded either when the subject reached 70 curl-ups, when the cadence was broken, or when their fingers did not reach the second line.

#### 2.4.4. The Sit-and-Reach Test

The subjects were in a sitting position on a flat floor with their legs fully extended. Their feet were placed against the base of the sit-and-reach box. Their toes pointed up, and they were aligned to the scale of 15 inches (38.1 cm). They were instructed to place one hand over the top of the other hand and bend slowly forward as far as possible. Each underwent the test three times, and the best score was recorded.

#### 2.4.5. The 300-Yard Shuttle Run

Subjects sprinted 50 yards, turned around, and sprinted 50 yards back. The process was performed three times for a total of 300 yards.

#### 2.4.6. 2400 Run

A standard runway of 400 m was selected, and the 2400-m run test started after the subjects heard the “Go” command. Subjects were asked to run as quickly as possible and the completion time was recorded.

### 2.5. The Combined Simulated Firefighting Tasks Procedures

The combined simulated firefighting procedures were presented in Figure 1. After warm-up and 15 min of rest, subjects performed the simulated firefighting tests. After 15 min of rest, physical performance tests were measured: pro-agility (5–10–5), one repetition maximum (bench press and deep squat), the crunch tests, the sit-and-reach test, the 300-yard shuttle run, and the 2400-m run. This procedure simulated a practical firefighting situation more closely. Heart rate was monitored using Polar Team (Polar Team2, Kempele, Finland) throughout the simulated firefighting tests, and fingertip capillary blood lactate concentrations were measured (EKF–C–Line, Ebendorfer, Germany) before and 3 min after the simulated firefighting tests. Urine samples were collected pre-simulated exercise in the morning and the first voided urine post-exercise. A total of 20 urine samples were collected, aliquoted, and stored at −80 °C, until proteomic analysis.

### 2.6. Urine Proteomics

#### 2.6.1. Sample Pretreatment

All samples were treated according to a previously published method [26]. Urine samples were removed from −80 °C storage and thawed at 4 °C, then centrifuged at 3000× *g* for 20 min at 4 °C to remove cells and debris. The sample was subsequently passed through a 15-mL Amicon ultrafiltration tube (Ultra-3k, Merck Millipore Ltd., Co. Cork, Ireland) and the flow-through was saved. Acetone (pre-chilled at −20 °C) was added to precipitate the proteins at −20 °C for 4 h. The precipitation was dissolved in 1% SDS/8M urea, and the protein concentration was determined using the BCA protein assay kit (Beyotime Biotechnology, Shanghai, China). For this assay, the maximum amount of SDS allowed in the protein sample without causing a noticeable interference is 5% [30]. The protein lysate was reduced in 10 mM of Dithiothreitol (DTT) at 37 °C, alkylated with 20 mM of iodoacetamide (IAM) at room temperature in darkness, and digested with trypsin at 37 °C. Digested peptides were desalted with a C18 spin column (Merck KGaA, Darmstadt, Germany). Peptide concentration was determined using a Peptide Assay, 500 Assays (Thermo Scientific, Waltham, MA, USA). The digested peptides were fractionated using the Pierce High pH Reversed-Phase Peptide Fractionation Kit (Thermo Scientific, Waltham, MA, USA). Fractions were pooled at various intervals to 10 fractions.

#### 2.6.2. LC–MS/MS Analysis

Liquid chromatographic (LC) analysis was performed using the Ultimate 3000 (Thermo Scientific, Waltham, MA, USA), which was equipped with an Acclaim PepMap RSLC C18 (75 μm × 150 mm, 2 μm, Thermo Scientific, Waltham, MA, USA). Mobile phase A was 0.1% formic acid and mobile phase B was 80% ACN/ 0.1% formic acid. After the determination of peptide concentration, for each sample, 9 μg of peptide was re-dissolved in 18 μL of solvent A (0.1% formic acid in water) and 2 μL 3× iRT, and then injected into LC–MS/MS. One injection per sample was conducted for all the tested samples. The resulting peptides were analyzed by Q Exactive^TM^ HF-X Orbitrap mass spectrometry system (Thermo Scientific, Waltham, MA, USA). The LC injection volume was 2 μL. The flow rate started at 0.8 μL/min at a temperature of 8 °C, decreased to 0.3 μL/min (after 4 min), and finally increased back to 0.8 μL/min at 87 min. The gradient condition was 0 min 1% B, 3.5 min linear gradient from 1% B to 5% B, 4 min linear gradient from 5% B to 6% B, 76 min linear gradient from 6% B to 28% B, 84 min linear gradient from 28% B to 37% B, 87 min linear gradient from 37% B to 99% B, hold 9% B until 90 min.

Data-dependent acquisition (DDA) was used to build a spectra library for data-independent acquisition (DIA) analysis. For DDA, the Thermo Q-Exactive HF-X mass spectrometer was operated in a positive polarity mode with a capillary voltage of 2 kV and a capillary temperature of 320 °C, and full MS-dd MS2 mode was used. MS1 scan properties were resolution 60,000, AGC target 3 × 10^6^, maximum injection time 50 ms, and scan range 350–1800 *m*/*z*. MS2 scan properties were a resolution of 15,000, an AGC target of 1 × 10^5^, a maximum injection time of 30 ms, TopN 20, and a collision energy of 28 eV. The sample was collected in DIA mode using a scan range of *m*/*z* 379–1205, a capillary voltage of +2 kv, a capillary temperature of 320 °C, a resolution setting of 30,000, an AGC target of 1 × 10^6^, and a collision energy of 25.5/27/30. DIA was performed with 60 variable isolation windows, and a full MS–SIM mode was applied to each of the 20 windows with a positive mode, a resolution setting of 120,000, an AGC target of 3 × 10^6^, a maximum injection time of 100 ms, and a scan range of 350–1250 *m*/*z*.

### 2.7. Statistical Analysis

To set up the spectral library, data of DDA data were processed and analyzed using default settings in Spectronaut13 (BiognosysAG, Switzerland) and then matched to the Homosapiens database (the UniProt Consortium, 2017). Data of DIA were also processed by Spectronaut13 under default settings and matched to the self-built spectral library and Homosapiens database to identify proteins. The retention time prediction type was set to dynamic iRT. The ‘Wu Kong’ platform (https://www.omicsolution.org/wkomics/main/, accessed on 15 May 2021) and UniProt (https://www.uniprot.org/, accessed on 15 May 2021) were used for protein identification and screening. Obtained protein ratios were log2 transformed. The significantly different proteins were determined by upregulated fold-change (FC) ≥ 2 or downregulated FC ≤ 0.5 in a Volcano map, and the adjusted *p* < 0.05 of a paired *t*-test. Pathway analysis was performed using Kyoto Encyclopedia of Genes and Genomes (KEGG, http://www.genome.jp/kegg, accessed on 20 May 2021), and Gene Ontology (GO) annotations (https://www.ebi.ac.uk/QuickGO/, accessed on 20 May 2021) were used for the functional annotation of proteins. Descriptive statistics were analyzed for the simulated firefighting test scores, heart rate, and blood lactate. Pearson correlation analysis was used to analyze the correlation between the physical performance, simulated firefighting test scores, heart rate, blood lactate, and urine proteomic data. All data were analyzed using SPSS 23.0 (Chicago, IL, USA), and the significant level was set as *p* < 0.05.

## 3. Results

### 3.1. The Results of Physical Performance Tests and Physiological Responses of the Combined Simulated Firefighting Tasks

Ten male firefighters were recruited into this study, and their demographic data were (mean ± standard deviation): age 24.5 ± 1.8 years, working years 5.2 ± 1.9 years, height 1.73 ± 0.5 m, weight 69.0 ± 4.9 kg, and body mass index (BMI) 23.0 ± 1.7 kg/m^2^.

All of the studied participants completed the five simulated firefighting tests with an average result of 174.20 ± 18.89 s. The descriptive data (mean and SD) for all physical performance scores are presented in Table 1.

The concentration of resting blood lactate was 2.16 ± 0.34 mmol/L, and after simulated rescue tasks, the average blood lactate concentration was 14.84 ± 2.08 mmol/L. During simulated rescue tasks, the mean heart rate was 170.10 ± 9.63 Bpm, and the maximum heart rate was 181.40 ± 6.45 Bpm. The concentration of protein present in urine before and after simulated firefighting was characterized using the BCA method (Figure 2). The protein concentration of fasting morning urine (0.093 ± 0.028 g/L) was normal (less than 15 mg/dL) [30], while, after the average protein concentration in the urine samples (0.325 ± 0.167g/L), post-simulated firefighting increased significantly, indicating proteinuria (urine protein 1 +) [31] (*p* = 0.001).

### 3.2. Changes of Urine Proteomics before and after the Combined Simulated Firefighting Tasks

A total of 1947 proteins were identified before and after the combined simulated firefighting tasks in this study. The principal component analysis (PCA) differentiation in the urine proteomics (Figure 3) was applied to discover the differences in proteins within and between sample groups (before and after the simulated firefighting). It showed that proteins identified were clustered into two discriminative groups obviously, indicating that proteins in the urine were significantly influenced by simulated firefighting. In order to further examine the proteomics changes, orthogonal partial least square discriminant analysis (OPLS-DA) was established to compare two groups’ urine samples (Figure 4), which showed that pre- and post-simulated firefighting match groups were in significantly different locations. The OPLS-DA model demonstrated good fitness (Q2 (cum) 0.924) and predictability (R2X (cum) 0.617, R2Y (cum) 0.987).

Volcano plots demonstrated that many proteins were up- or down-regulated (Figure 5) before simulated firefighting compared to after simulated firefighting. There are 625 significantly differential proteins between the before and after groups, including 360 proteins with significant upregulation and 265 proteins with significant downregulation (fold change (FC) of >2 or <0.5, and a *p*-value of <0.05).

### 3.3. Pathway Enrichment Analysis

The pathway analysis could elucidate the significant pathways, including the signal transduction pathway that the differential proteins altered by simulated firefighting were involved in. The pathway analysis demonstrated that complement and coagulation cascades were the top pathways involved with the lowest *p*-value. Other following pathways were lysosome, cell adhesion molecules, ECM–receptor interaction, PPAR signaling pathway, amoebiasis, cholesterol metabolism, and glycosaminoglycan degradation (Figure 6).

### 3.4. GO Enrichments

Gene Ontology (GO) enrichments for the proteins with significant variations are displayed in Figure 7. The annotation results revealed biological processes (top three: cell adhesion, complement activation classical pathway, and signal transduction), cellular composition (top three: extracellular exosome, plasma membrane, and extracellular region), and molecular functions (top three: calcium ion binding, serine-type endopeptidase activity and antigen binding).

### 3.5. Correlations between the Physical Performance and Simulated Firefighting Tests

Firefighting rescues require high levels of aerobic fitness, anaerobic capacity, muscular strength, endurance, and so on [6,14,15]. Some simple and effective physical performance tests could reflect physical fitness such as agility [32], strength [33], endurance [34], flexibility [35], anaerobic [36], and aerobic fitness [37]. Therefore, this study performed correlation analysis between the physical performance and simulated fire-fighting tests. The results indicated that the crunch tests score was negatively correlated with the simulated firefighting testing completion time (*r* = −0.671, *p* = 0.034), and the time of the 300-yard shuttle run was positively correlated with the simulated firefighting testing completion time (*r* = 0.731, *p* = 0.016). These results indicated that endurance and anaerobic fitness were significant to the simulated firefighting performance.

### 3.6. Correlations between the Urine Proteomics Changes and Simulated Firefighting Performance

Blood lactate and heart rate are ideal indicators to reflect firefighter’s physical condition and exercise load [16]. We performed correlation analysis between significant proteins (top 20 upregulated and top 20 downregulated proteins) and physiological indicators (Figure 8).

Posttest blood lactate was positively correlated with carbonic anhydrase 1, but was negatively correlated with activin receptor type-1B, plexin domain-containing protein 2, plexin-A2, multiple epidermal growth factor-like domains protein 8, small integral membrane protein 5, and natural cytotoxicity triggering receptor 3 ligand 1. The mean heart rate during simulated firefighting was positively correlated with the beta 2-microglobulin, procollagen C-endopeptidase enhancer 2, IGF-like family receptor 1, CD320 antigen, von Willebrand factor, collagen alpha-1(XII) chain, and carcinoembryonic antigen-related cell adhesion molecule 8. We also found that, after simulated firefighting, the protein concentration in the urine was related to the beta 2-microglobulin (*r* = 0.659, *p* = 0.038).

## 4. Discussion

The aim of this study was to explore the characteristics of urinary proteomics of the combined simulated firefighting test, and to find ideal biomarkers to assess the related physiological responses. The results indicated that simulated firefighting tests induced significant urinary proteomic changes, and there were 360 proteins significantly up-regulated and 265 proteins significantly downregulated after this simulated test. To our knowledge, this was the first study to employ proteomics technology to analyze urine proteomics of simulated firefighting tasks. The results also suggested that beta 2-microglobulin, von Willebrand factor, and carbonic anhydrase 1 were strongly associated with the physiological responses to simulated firefighting tests, which could be used as biomarkers to monitor firefighters.

The simulated firefighting in this study involved high-intensity exercise, which well reflected the intensity of actual rescues. The exercise terms in this study required agility, muscular strength, flexibility, aerobic capacity, and anaerobic capacity, which was closer to the complicated real rescue. When the firefighters were actively conducting simulated firefighting, the mean heart rate reached 170.10 ± 9.63 bpm ((89.79 ± 0.053%) HRmax (206.9 − 0.67 × age)), which was consistent with previous work [16,17]. The blood lactate value pre- and post-simulated firefighting rose from 2.16 ± 0.34 mmol/L to 14.84 ± 2.08 mmol/L (*p* < 0.05). Dennison [31,38] also reported that the post-test blood lactate value was 11.65 ± 2.78 mmol/L in trained firefighters. The simulated firefighting significantly increased the concentrations of urinary proteins from 0.093 ± 0.0275 g/L to 0.325 ± 0.1665 g/L (Urine protein 1 +: urine protein concentrations are larger than 0.3 g/L [31]) (*p* = 0.001), and the protein concentration was calculated using the BCA assay. Papassotiriou [29] reported that, after a typical session of the preparatory phase, proteinuria, approximately 30 mg/L, was found in 60% of athletes applying the relevant assay for microprotein concentrations. The main cause of exercise-induced proteinuria is not clarified, but the reninangiotensin system and prostaglandins might play a significant role [28]. The plasma angiotensin II concentration increases at intense physical activity, possibly leading to filtration of proteins through the glomerular membrane [28]. Moreover, in this study, the proteomic analysis indicated that angiotensinogen and beta 2-microglobulin were also significantly increased in urine post-simulated exercise, which were related with glomerular leakage [39,40]. These physiologic changes suggested that the intensity of the simulated firefighting was high.

Based on the proteomic results, the simulated firefighting was an aerobic-dominant exercise, and the oxidation of carbohydrates and fatty acid was the main energy source. The tricarboxylic acid cycle was intensified, reflected by the elevation of cytosolic phosphoenolpyruvate carboxykinase (PCK1). The research has shown that, at high glucose, PCK1 acetylation by p300 promotes PCK1 anaplerotic activity (PEP/OAA), and synthesized OAA from PEP can occur in vivo, urning the gluconeo genic pathway kinetically unfavorable and replenishing the TCA cycle [41]. There was also an increased level of adiponectin after the simulated firefighting. Adiponectin is an important hormone secreted by adipocytes, which stimulates glucose utilization and fatty acid oxidation [42]. Moreover, zinc-a2-glycoprotein (ZAG) was also significantly increased in urine post-exercise. Previous studies have shown that ZAG is an adipokine involved in stimulating adipocytes to induce lipid degradation and stimulates adiponectin release from human adipocytes [43]. The KEGG analysis also indicated that the PPAR signaling pathway, metabolic regulators of adipocyte differentiation, and glucose and lipid metabolism were up–regulated. Additionally, the 6-phosphogluconate dehydrogenase, decarboxylating (G6PD), was significantly decreased in urine post-exercise. The overall reduced oxidative pentose phosphate pathway (oxiPPP) flux in G6PD, 6-phosphogluconolactonase (PGLS), or 6-phosphogluconate dehydrogenase (6PGD) knockdown cells might redirect glucose-6-phosphate to glycolysis, causing increased glycolytic rate and ATP levels [44]. These results all suggested that the energy of this simulated firefighting was mainly provided by aerobic metabolism of carbohydrates and lipids. By GO functional enrichment analyses and KEGG analysis, we found that simulated firefighting could provoke acute inflammatory responses, immune responses, complement activation, and oxidative stress. The protein expression of the inflammation-associated factors, i.e., NF-κB, TNF-α, IL-6, and IL-1β increased simultaneously. Specifically, alpha-1-acid glycoprotein 1 (ORM1), alpha-1-acid glycoprotein 2 (ORM2), interleukin-1 receptor accessory protein (IL1RAP), and tumor necrosis factor receptor superfamily member (TNFRSF) 1A were up-regulated, and interleukin-18-binding protein (IL-18BP) were down-regulated, which indicated that simulated firefighting caused slightly up-regulated inflammation effects [45]. This study indicated that there were elevations in immunoglobulin light and heavy chains, and the up–regulated complement C4-A and C4-B may result in the activation of complement pathways, exacerbating the inflammatory response [46]. The KEGG pathway analysis indicated that the complement and coagulation cascades pathway was significantly up-regulated. Peroxiredoxin-5 (PRDX5), mitochondrial, superoxide dismutase [Cu–Zn] (SOD1), and glutathione peroxidase 3 (GPX3) all increased in urine post-simulated firefighting, which indicated that simulated firefighting led to oxidative stress response [47,48,49].

The simulated firefighting tasks were aerobic-dominant high-intensity exercise, while there were some differences between the tasks and other aerobic-dominant high-intensity sports. Regular aerobic exercise, such as marathon and high-intensity interval training (HIIT), is a regular periodic exercise [26,50]. Shi et al. [50] explored the metabolomics characteristics of serum before and after a marathon. The results indicated that the carbohydrates, proteolysis, and amino acids were all elevated to provide the energy for the long-term exercise. Zhao et al. [26] revealed the metabolic characteristics of HIIT by metabolomics and proteomics methods. They found 51 upregulated proteins and 51 downregulated post-exercise compared to pre-exercise. HIIT could provoke changes in protein expression associated with energy production, immune function, and metabolism. In terms of energy production, most of the differential proteins were related to carbohydrate metabolism, lipid metabolism, and amino acid metabolism. After HIIT exercise, the biomarkers of muscle damage were not increased.

By further correlations analysis, we found that some proteins could be regarded as important metabolic indicators to reflect the level of metabolic state and physiological responses for simulated firefighting. Heart rate is often measured to evaluate exercise intensity [51]. Beta-2-microglobulin (*r* = 0.76, *p* < 0.05) and von Willebrand factor (vWF) (*r* = 0.81, *p* < 0.01) were positively correlated with heart rate. The serum beta-2-microglobulin concentration has been proved to be an independent predictor of major cardiovascular events and total mortality in the general population or elderly patients whose glomerular filtration rate (eGFR) was in normal range [52]. Exercise could increase the release of vWF by sympathetic activation, and vWF has been proven to be associated with cardiovascular diseases, particularly exercise-induced pathological outcomes [53]. Therefore, beta-2-microglobulin and vWF may be regarded as a significant indicator to monitor exercise intensity and cardiovascular risks for firefighting. Blood lactate is closely related to exercise intensity and is one of the significant indicators for judging the exercise endurance and fatigue resistance [54,55]. In this study, carbonic anhydrase 1 (*r* =0.67, *p* < 0.05) was positively correlated with blood lactate. It was reported that the immobilized enzyme carbonic anhydrase (CA) could increase the carbon dioxide diffusional gradient in blood [56]. Moreover, inhibition of CA is related to a lower plasma lactate concentration during fatiguing exercise [57]. Therefore, CA 1 could be used as a significant indicator to evaluate exercise endurance and fatigue resistance for firefighters.

For subjects of other sports, there are also some correlations between the omics characteristics of urine or serum and physiological parameters. Yang et al. [58] investigated serum proteomic markers in individuals participating in 12 weeks of tai chi chuan (TCC) exercise. They reported that complement factor H may represent a useful marker of the health effects of exercise. Ali et al. [59] recruited ten physically active healthy adults to perform two cycling exercise trials, and urine samples were collected for metabolomic analysis. They found that oxo-aminohexanoic acid (OHA) was strongly correlated with VO2max (R2 = 0.8589) which could serve as a potential predictive marker for VO2max. Shi et al. [50] recruited 20 adult males who participated in a marathon, and then explored metabolomic changes of serum following the marathon. They found that serum metabolites (cis-aconitate, galactonic acid, and mesaconicacid) were closely related to VO2max (*r* = −0.679, −0.474, −0.708). Therefore, it is reasonable to speculate that the correlation between high-intensity physical activity and the omics profile is more possibly dependent on the type and intensity of exercise than on occupational characteristics, such as firefighting.

There are some limitations of this study. First, the study sample sizes were relatively small, and there was slight interindividual variability. Second, continuous monitoring of urinary omics characteristics after simulated firefighting tasks was lacking. In the future, the sample size could be increased. We could also perform continuous monitoring of the characteristics of urinary omics. To completely present the firefighting-induced physiological responses, serum samples could also be collected for omics analysis.

## 5. Conclusions

The simulated firefighting tasks included an aerobic-dominant high-intensity exercise, which promoted the catabolism of carbohydrates and lipids. The simulated firefighting could provoke acute inflammatory responses, immune responses, complement activation, and oxidative stress. Beta-2-microglobulin and von Willebrand factor could reflect exercise intensity and cardiovascular risks of firefighting, and carbonic anhydrase 1 could be regarded as an important indicator to evaluate exercise endurance and fatigue resistance for firefighters, but further research is needed for confirmation. More research is needed to confirm that whether these biomarkers are specific for firefighters or also suitable for other professions.

## Figures and Tables

**Figure 1 ijerph-18-10618-f001:**
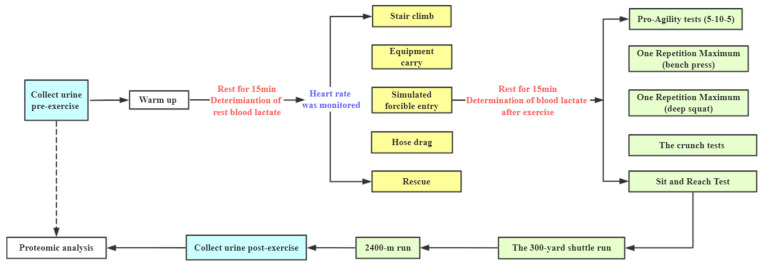
Flowchart of the combined simulated firefighting tasks procedures.

**Figure 2 ijerph-18-10618-f002:**
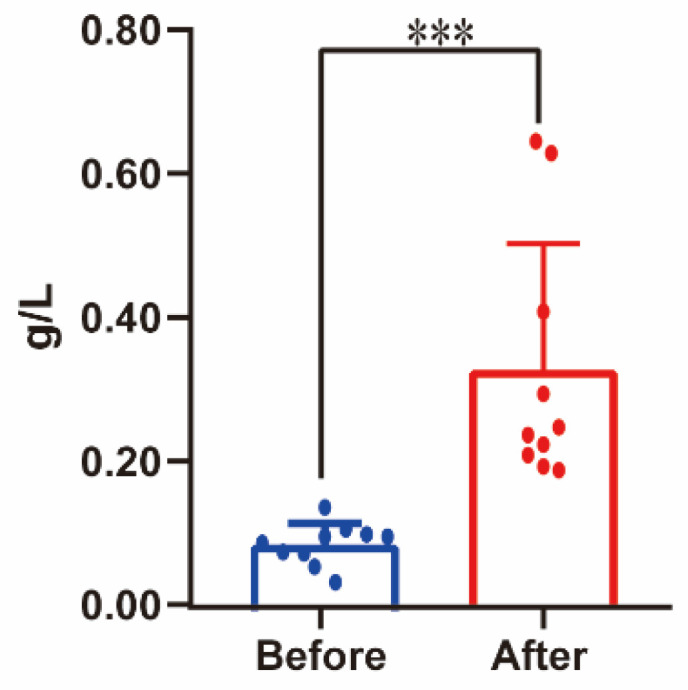
The concentration of protein present in urine before and after the combined simulated firefighting tests. *** *p* < 0.01.

**Figure 3 ijerph-18-10618-f003:**
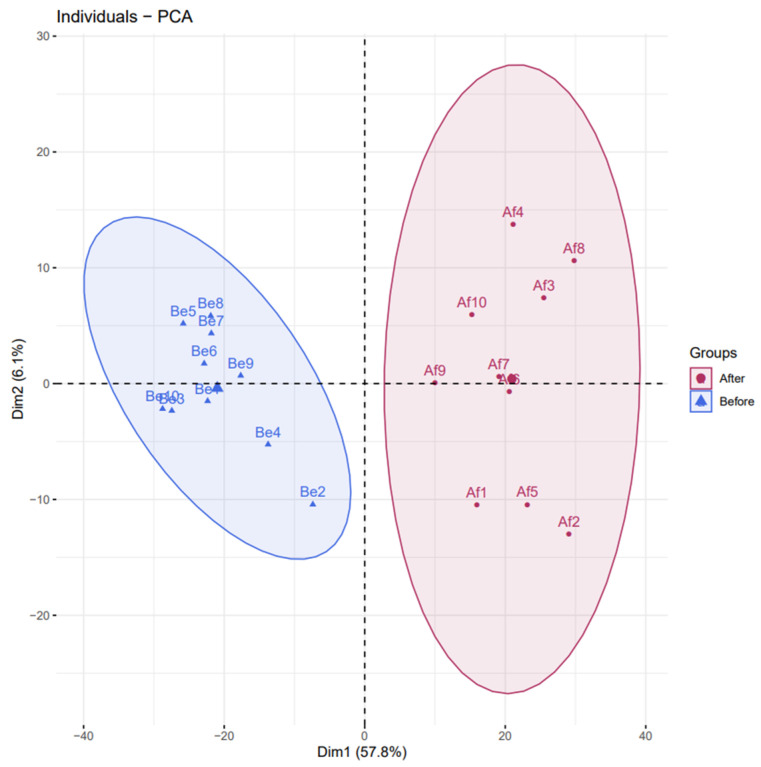
PCA plot for proteomics datasets of urine samples collected before the combined simulated firefighting test and after this.

**Figure 4 ijerph-18-10618-f004:**
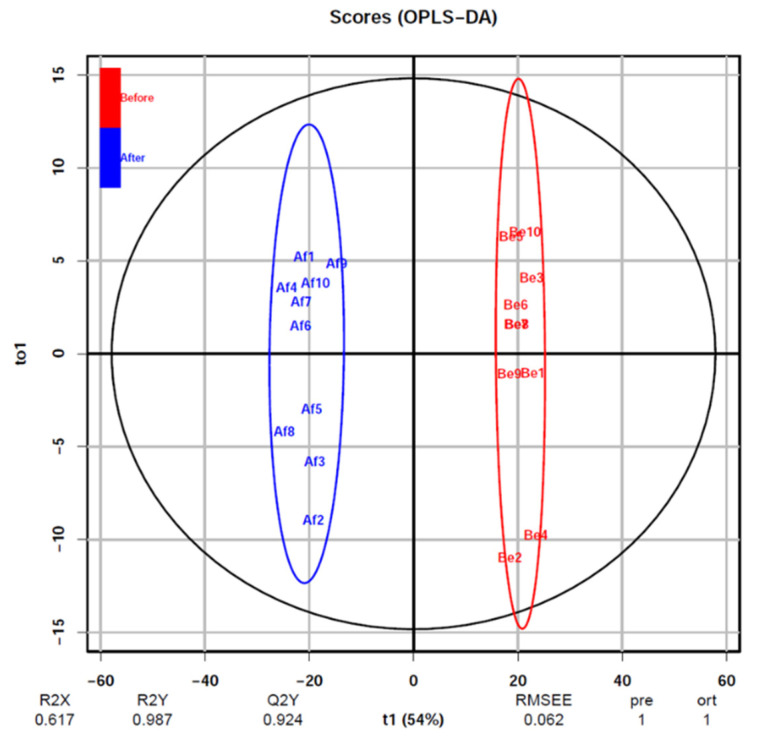
OPLS-DA plot for proteomics datasets of urine samples collected before the combined simulated firefighting test and after this.

**Figure 5 ijerph-18-10618-f005:**
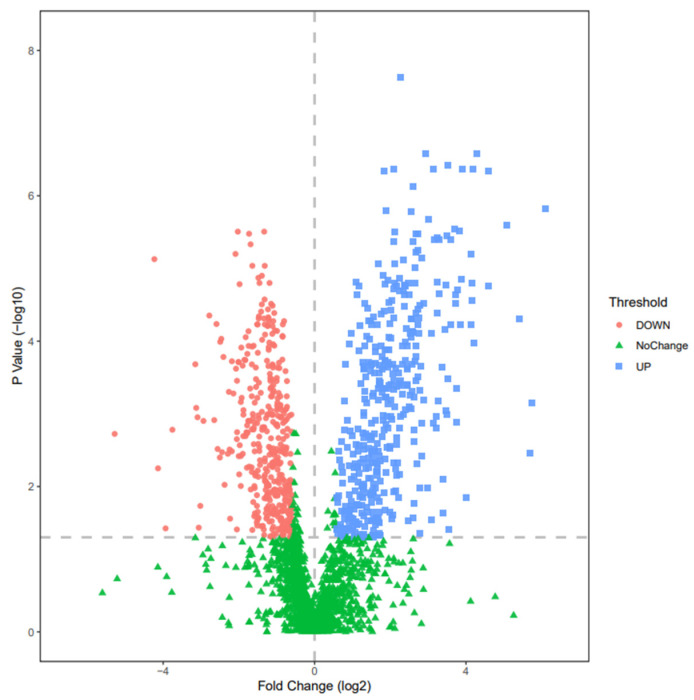
Volcano plot for proteomics datasets of urine samples collected before the combined simulated firefighting test and after this.

**Figure 6 ijerph-18-10618-f006:**
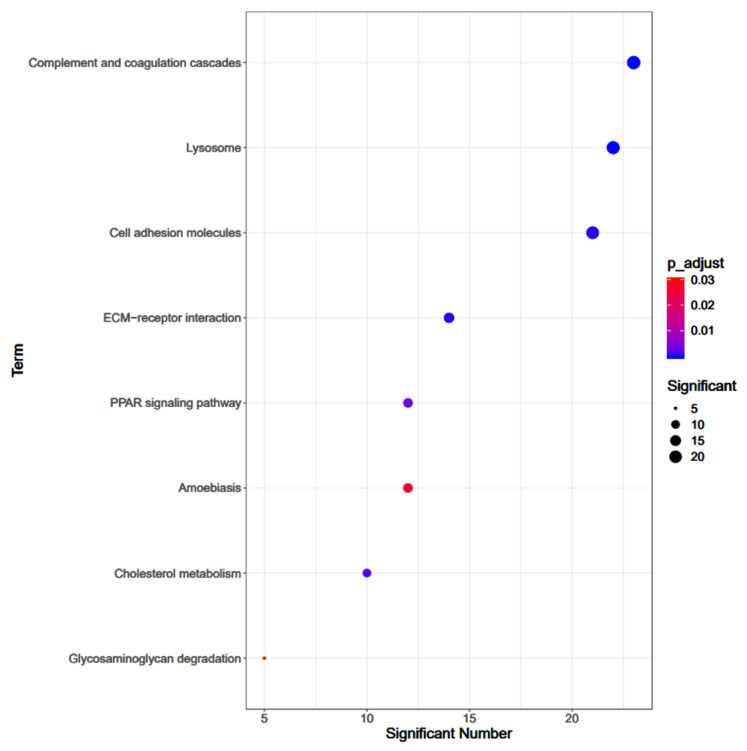
The KEGG pathway analysis of the differentially expressed proteins.

**Figure 7 ijerph-18-10618-f007:**
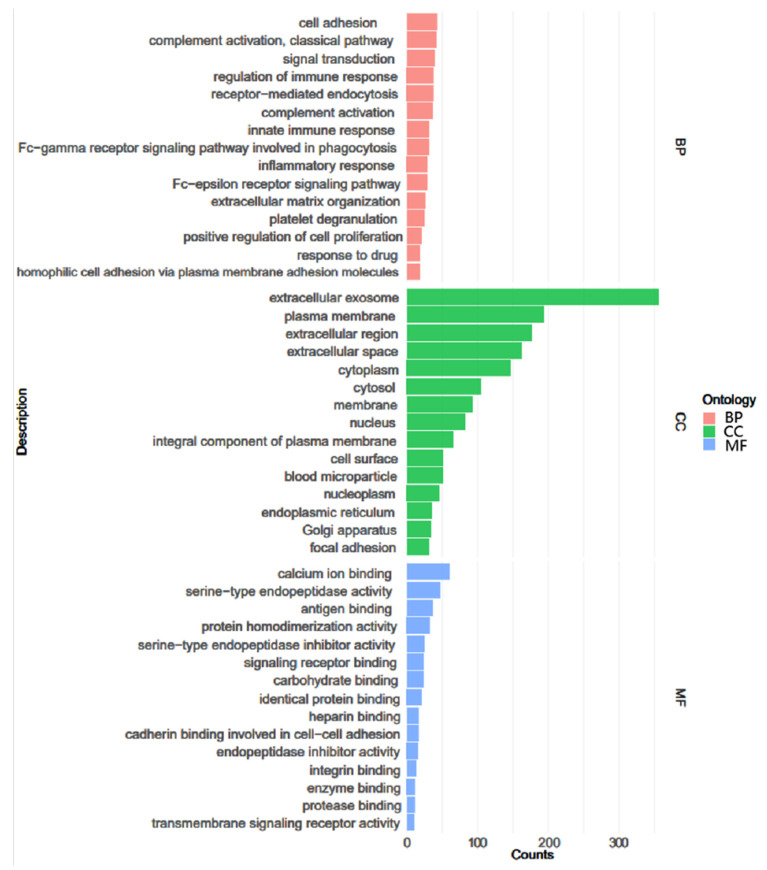
Gene ontology (GO) annotation of differential proteins, which were divided into 3 categories: biological process (BP), cellular component (CC), and molecular function (MF). The top 15 components for BP, CC, and MF of the differential proteins according to the GO database are shown.

**Figure 8 ijerph-18-10618-f008:**
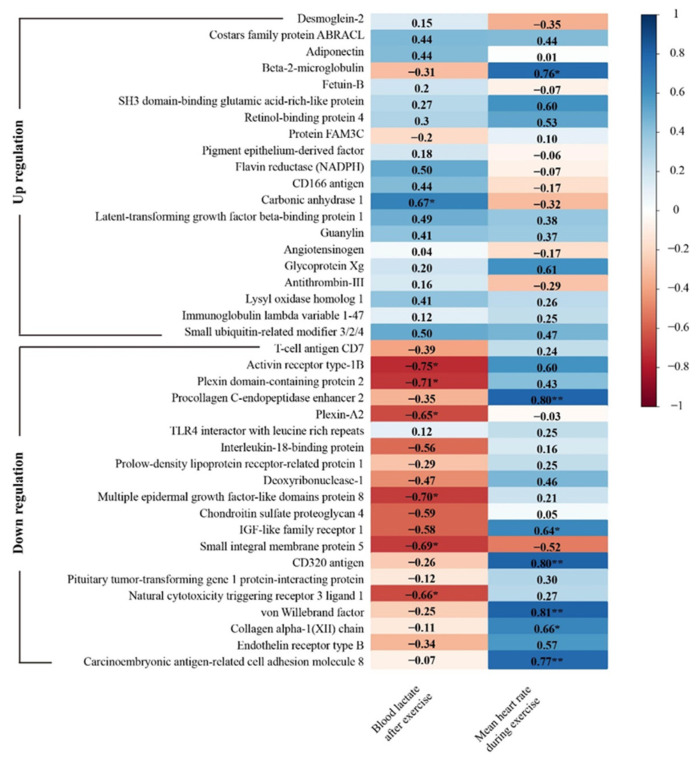
Correlations between the urine proteomics changes and simulated firefighting performance. * *p* < 0.05, ** *p* < 0.01.

**Table 1 ijerph-18-10618-t001:** The results of each physical performance test (*n* = 10).

Pro–Agility (s)	1 RM (Bench Press) (kg)	1 RM (Deep Squat) (kg)	The Crunch Tests (a.u)	Sit-and-Reach (cm)	The 300-Yard Shuttle Run (s)	2400-m Run (s)
4.94 ± 0.21	121.92 ± 15.59	80.21 ± 11.64	45.00 ± 16.78	43.75 ± 8.16	63.28 ± 3.01	571.87 ± 31.44

One Repetition Maximum; a.u: arbitrary units.

## Data Availability

Data is contained within the article or Appendix A.

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
