# Peer review of "Urinary Proteomics of Simulated Firefighting Tasks and Its Relation to Fitness Parameters"

_ijerph, 2021, doi:10.3390/ijerph182010618_

Round 1

Reviewer 1 Report

Knowledge of the metabolic characteristics during realistic firefighting rescue is of great value for firefighters to design reasonable recovery programs and to reduce their rate of injury. Although some indexes including the heart rate, oxygen uptake, expiratory ventilation, blood lactate concentration and rating of perceived exertion were measured to some specific physiological alterations in simulated firefighting tasks, it is difficult to reflect the effect of simulated rescue on physiology in the molecular scale. The present study first time explored the simulated tasks-induced physiological responses using the urine proteomics analysis. Additionally, some biomarkers were found. In my opinion, this manuscript could be accepted for publication. However, the following suggestions should be considered.

  1. The study concluded that the simulated firefighting tasks were an aerobic-dominant high intensity exercise, which promoted the catabolism of carbohydrates and lipids. What are the differences in the aerobic-dominant high intensity exercise between this type of physical sports and other sports?
  2. If there is a general association between high-intensity physical exercise and proteomic metrics, whether the results of this study are consistent with that of previous studies. Namely, these results mainly depend on the high-intensity physical exercise rather than feature of profession, and there are some certain relations between the indexes of proteomic metrics and the physiological indexes of physical exercise, and this relationship is not necessarily related to the profession. Authors should consider the suggestions mentioned above in the discussion and conclusion section.
  3. In 2.3 Simulated firefighting tasks, these tasks should be compared with that reported in literature, reflecting their reasonability or consensus.

      4. In section of 2.6 Urine proteomics, some references should be cited.            In  section of “3.5 Correlations between the physical performance and            simulated firefighting tests” , there is a similar matter. 

Reviewer 2 Report

Zhu et al describes the proteomic urine profiling pre and after a simulation exercise program of firefighters. The experimental approach is well controlled.

The urine samples are processed by ultracentrifugation: please describe this in more detail (types of filters used?). Also, are the urine samples positive for the presence of cells?

How severe is the proteinurea? How long is the proteinurea lasting? What is the interindividual variability? How was the SDS presence affecting the BCA measurements?

Is this also reflected in the serum? as an increase of creatinine? 

The MS method for the DDA and DIA measurements should be included in the manuscript.

The number of replica measurements (if performed) should be clearly stated.

How much peptide amount was injected, was it the same for each sample?

The complete protein list should be provided as supplementary table together with the peptides and their quantification data.
